# Overexpression of *PagERF072* from Poplar Improves Salt Tolerance

**DOI:** 10.3390/ijms231810707

**Published:** 2022-09-14

**Authors:** Xuemei Zhang, Zihan Cheng, Wenjing Yao, Yuan Gao, Gaofeng Fan, Qing Guo, Boru Zhou, Tingbo Jiang

**Affiliations:** 1State Key Laboratory of Tree Genetics and Breeding, Northeast Forestry University, Harbin 150040, China; 2Bamboo Research Institute, Nanjing Forestry University, 159 Longpan Road, Nanjing 210037, China

**Keywords:** poplar, *PagERF072*, genetic transformation, salt stress

## Abstract

Extreme environments, especially drought and high salt conditions, seriously affect plant growth and development. Ethylene-responsive factor (ERF) transcription factors play an important role in salt stress response. In this study, a significantly upregulated ERF gene was identified in 84K (*Populus alba* × *P. glandulosa*), which was named *PagERF072*. *PagERF072* was confirmed to be a nuclear-localized protein. The results of yeast two-hybrid (Y2H) assay showed that PagERF072 protein exhibited no self-activating activity, and yeast one-hybrid (Y1H) demonstrated that PagERF072 could specifically bind to GCC-box element. Under salt stress, the transgenic poplar lines overexpressing *PagERF072* showed improved salt tolerance. The activities of peroxidase (POD), superoxide dismutase (SOD) and catalase (CAT) in transgenic poplars were significantly increased relative to those of wild-type (WT) plants, whereas malondialdehyde (MDA) content showed an opposite trend. In addition, reactive oxygen species (ROS) was significantly reduced, and the expression levels of POD- and SOD-related genes were significantly increased in transgenic poplars under salt stress compared with WT. All results indicate that overexpression of the *PagERF072* gene can improve the salt tolerance of transgenic poplars.

## 1. Introduction

Abiotic stresses, especially high salt conditions, seriously affect plant growth and development. Under salt stress, the dynamic balance of ions in plants is considerably damaged, which causes membrane damage and cell death [1,2]. The cell membrane system of plants can be easily damaged, which can interfere with the balance between the production and removal of reactive oxygen species (ROS) in cells, causing plants to suffer from oxidative stresses [3,4]. Similarly, the photosynthesis system or physiological metabolism of plants is disturbed by salt stress [5,6].

Transcription factors (TFs) play important roles in plant growth and stress response. TFs can specifically bind to *cis*-acting elements in the promoters of target genes to regulate their spatiotemporal expression, responding to external environmental stresses [7,8]. APETALA2/ethylene responsive factor (AP2/ERF) family is one of the largest plant-specific TFs families and contains at least one conserved AP2 DNA binding domain consisting of approximately 60 amino acids. The first AP2/ERF family gene was discovered in Arabidopsis and named *APETALA2*; it contains two AP2 domains and plays an important role in flower development [9]. AP2/ERF family genes were subsequently reported in radish [10], adzuki bean [11], barley [12] and other species. The AP2/ERF family can be divided into AP2, ERF, dehydrate-responsive element binding factor (DREB), RAV and Soloist subfamilies [13]. The ERF subfamily contains a single AP2 domain, and the amino acids located at 14th and 19th positions are alanine and aspartic, respectively. Researchers have reported that ERF family genes are involved in plant development, including fruit development [14], seed dormancy [15], flower development [16] and plant growth [17].

ERF transcription factors are involved in plant metabolic regulation. For example, the *MdERF38* gene interacts with an anthocyanin-positive regulatory gene, *MdMYB1*, to increase anthocyanin content in apple [18]. Transient expression of *AaERF1* or *AaERF2* in tobacco increases transcription levels of *ADS* and *CYP71AV1*, leading to accumulation of artemisinin and artemisinic acid [19]. Moreover, ERF genes play important roles in response to abiotic stresses. For example, Arabidopsis gene *AtERF98* promotes transcriptional activation of ascorbic acid synthesis to enhance salt tolerance [20]. Rice gene *OsERF48* directly controls its downstream target, the *OsCML16* gene, to enhance drought tolerance [21]. Soybean gene *DREB1* interacts with *GmERF008* and *GmERF106* to regulate drought tolerance [22]. In previous research, we found that overexpression of *ERF38* and *ERF76* can increase salt tolerance of transgenic poplars [23,24].

84K (*Populus alba* × *P. glandulosa*) is derived from the hybrid offspring of *Populus alba* and *Populus glandulosa*. It has the characteristics of fast growth, a well-developed root system, good material quality and easy survival from cuttings. Therefore, it has become an important model plant in forest tree genetics and breeding research. In this study, we found a highly salt-induced *PagERF072* gene under salt stress [25]. The expression pattern of the *PagERF072* gene was validated by qRT-PCR. The gene was cloned and transformed into poplar using an *Agrobacteria*-mediated method. The growth phenotypes of transgenic poplars were observed, and physiological indices were measured under salt stress. In this study, we identify the molecular function of the *PagERF072* gene, potentially contributing to understanding of the mechanism of salt resistance in poplar.

## 2. Results

### 2.1. Screening and Cloning of the ERF072 Gene

We obtained an mRNA abundance of 174 ERF family genes from previous research [25] (Appendix A) and found 35 significant differentially expressed genes (DEGs), including 16 upregulated genes and 19 downregulated genes (Figure 1A). In addition, we found that *PagERF072* had a higher expression level under salt stress (Figure 1B). Subsequently, we successfully cloned the *PagERF072* gene from 84K (*Populus alba* × *P. glandulosa*), and the full-length sequence of the *PagERF072* gene was 747 bp, encoding 248 amino acid residues.

### 2.2. Sequence Alignment and Phylogenetic Analysis of PagERF072 Protein

Multiple sequences alignment indicated that PagERF072 protein has an AP2 DNA binding domain (Figure 2A). The amino acid sequence similarity of PagERF072 and its homologous proteins from *Populus trichocarpa*, *Salix brachista*, *Durio zibethinus*, *Gossypium mustelinum*, *Vernicia fordii*, *Manihot esculenta*, *Hibiscus syriacus*, *Pistacia vera*, *Corchorus capsularis*, *Jatropha curcas* and *Arabidopsis thaliana* was 93.95%, 72.37%, 51.64%, 50.55%, 53.88%, 51.10%, 50.57%, 52.85%, 50.54%, 53.18% and 43.12%, respectively. Phylogenetic analysis indicated that PagERF072 was closely related to XP 002315490.2 from *Populus trichocarpa* (Figure 2B).

### 2.3. Gene Expression Pattern Analysis of PagERF072

We then identified the expression pattern of *PagERF072* in different tissues by qRT-PCR. As shown in Figure 3A, the expression level of the *PagERF072* gene was high in the primary stems and secondary leaves, whereas it was relatively lower in the secondary stems and roots. In addition, the expression levels of *PagERF072* in the leaves and roots were measured at different time points under salt stress and abscisic acid (ABA) treatments (Figure 3B), respectively. The expression pattern was similar in these two tissues. Under salt stress, the expression level of *PagERF072* first increased and reached its peak after 24 h, at which point it decreased. However, *PagERF072* had the highest expression level at 6 h under ABA treatment.

### 2.4. PagERF072 Protein Was Localized in Nucleus and Exhibited No Activation Activity

In order to confirm the localization of PagERF072 protein, pBI121-ERF072-GFP and pBI121-GFP plasmids were transformed into onion epidermal cells by particle bombardment. As shown in Figure 4A, the fluorescent signal of the control vector was located in the nucleus and cytoplasm, whereas the GFP signal only existed in the nucleus of epidermal cells when transformed with a pBI121-ERF072 fusion vector. The results confirmed that PagERF072 was localized in the nucleus.

A yeast two-hybrid (Y2H) assay was performed to verify the transcriptional activity of PagERF072 protein. As shown in Figure 4B, the positive control (pGBKT7-53/pGADT7-T), negative control (pGBKT7) and ERF072-BD yeast strains grew normally on SD/-Trp medium. However, only the positive control yeast strain grew and turned blue on SD/-Trp/-his/X-α-Gal, which indicates that PagERF072 protein exhibited no transcriptional activation activity.

### 2.5. PagERF072 Specifically Binds to GCC-Box Element

A yeast one-hybrid (Y1H) system was used to determine whether PagERF072 could bind to GCC-box element. As shown in Figure 5A, the positive control, negative control and PagERF072 yeast strains grew normally on SD/-Ura medium. Both the PagERF072 yeast strain and the positive control grew normally on the SD/-Leu/AbA medium, whereas the negative control did not grow, which indicates that PagERF072 can bind to GCC-box elements.

In addition, we carried out transient transformation of *PagERF072* into tobacco to further verify the above results. When the protein specifically bonded to GCC-box element, the 35S promoter of the effector vector was able to drive *GUS* expression in the reporter vector (Figure 5B). As shown in Figure 5C, when pROK-ERF072 and GCC-box strains were cotransformed into tobacco, the leaves of transgenic tobacco became blue after staining, whereas the leaves of control plants were not stained. All the results show that PagERF072 protein can specifically bind to GCC elements.

### 2.6. Molecular Identification of Transgenic Plants

Three positive transgenic poplar lines were identified by PCR with specific primers. The exogenous *PagERF072* fragment was only detected in transgenic poplars (Figure 6A). Moreover, transgenic poplars took root normally in the rooting medium containing 50 mg/L kanamycin, whereas WT poplars did not root (Figure 6B). Then, the expression levels of the *PagERF072* gene were quantified in the transgenic poplar lines. The results show the transgenic lines had significantly higher expression levels than WT plants (Figure 6C).

### 2.7. Overexpression of PagERF072 Increases Salt Tolerance of Transgenic Poplar

To test the salt tolerance of transgenic poplar lines, seedlings were irrigated with 150 mM NaCl for five days. As shown in Figure 7A, no obvious growth difference was observed between transgenic poplars and WT plants under normal conditions. Under salt stress, the leaves of WT poplars were severely dehydrated and wilted. However, the leaves of transgenic lines maintained a relatively positive condition. The result indicates that the *PagERF072* gene improved the salt tolerance of transgenic poplars.

We also measured some physiological indicators of transgenic poplars under salt stress. As shown in Figure 7B, under control conditions, no apparent physiological difference was observed between transgenic poplar lines and WT plants. However, under salt stress, the activities of POD, SOD and CAT in transgenic poplars were significantly higher than those of WT plants, whereas MDA content was significantly reduced in transgenic poplars relative to WT plants (Figure 7B).

### 2.8. Overexpression of PagERF072 Enhances ROS Scavenging in Transgenic Poplar

NBT and DAB staining were conducted to detect the content of hydrogen peroxide and superoxide in transgenic poplars under salt stress. As shown in Figure 8A, under normal conditions, no apparent difference in staining degree was observed between transgenic lines and WT plants. Under salt stress, the staining color of WT leaves was much deeper than that of transgenic leaves.

Similarly, the expression levels of POD- and SOD-related genes were determined by RT-qPCR. As shown in Figure 8B, under the control conditions, the expression levels of POD- and SOD-related genes did not differ significantly between transgenic lines and WT. Under salt stress, the expression levels of POD- and SOD-related genes in transgenic lines were significantly higher compared with those of WT plants, indicating that *PagERF072* may regulate the expression of the POD- and SOD-related genes, improving plant resistance.

## 3. Discussion

The AP2/ERF family is major plant-specific TF family containing several subfamilies, including AP2, RAV, DREB, ERF and Soloist subfamilies [13]. Genome-wide analysis of the ERF subfamily has been reported in Arabidopsis [26], poplar [27], adzuki bean [11], Barley [12], etc. In previous studies, Arabidopsis *ERF072* was reported to participate in anaerobic stress response [28]. In addition, the *ERF072* gene is involved in iron deficiency [29] and plays an important role in responses to low temperature, oxidation and osmotic stress [30]. In this study, we identified a highly salt-induced ERF gene, *PagERF072*, which is a homologous gene of Arabidopsis *ERF072*. The gene encodes 248 amino acids and contains a unique AP2 DNA-conserved domain. Subcellular localization indicates that PagERF072 protein is localized in the nucleus. Evidence from a yeast hybrid system indicates that PagERF072 protein has no transcriptional activation activity and can specifically bind to GCC-box element.

The ABA signaling pathway plays an important role in responding to abiotic stress [31]. Numerous studies have suggested that many ERF family genes act as ABA signaling messengers in response to abiotic stress in plants. For example, Arabidopsis ERF gene *ORA47* gene coordinates with two ABA-related genes, *ABI1* and *ABI2*, to form an ABI1–ORA47–ABI2 positive feedback loop to respond to emergency stress [32]. Overexpression of potato ERF gene *IbRAP2-12* could increase the expression of ABA-related genes to increase salt and drought tolerance of transgenic plants [33]. In this study, we found that *PagERF072* was highly expressed under ABA stress. Therefore, we speculated that *PagERF072* may participate in the ABA signaling pathway to respond to salt stress.

ERF family genes often display specific gene function under stresses. For example, ERF gene *ORA59* interacts with *RAP2.3* to jointly regulate ethylene sensitivity and disease tolerance [34]. Overexpression of *RAP2.12* and its homologues, *RAP2.2* and *RAP2.3*, activates hypoxia-marked genes under hypoxic and normoxic conditions [30]. Inducible expression of all three *RAP2*s confers tolerance to hypoxia, as well as oxidative and osmotic stresses [30]. In addition, many AP2/ERF genes, such as *TaERF3* [35], *IbRAP2-12* [33] and *ThERF1* [36], have also been found to function in plant responses to salt stress. In this study, we obtained three transgenic poplar lines overexpressing *PagERF072*. The transgenic poplars displayed growth advantages relative to WT plants under salt stress, whereas no significant difference was observed under control conditions. The results indicate that *PagERF072* plays an important role in improving salt tolerance.

Under salt stress, a large amount of ROS is produced in plants, which causes damage of the membrane structure, lipids, carbohydrates and protein structure [37,38]. POD and SOD act as antioxidant proteins, providing the main defense against various stress conditions, which can effectively reduce ROS accumulation in plants when exposed to stresses [39,40,41]. Numerous studies have proven that ERF genes can regulate antioxidant enzyme genes to respond to abiotic stresses. For example, overexpression of the *Larix olgensis LoERF017* gene can increase the activity of antioxidant enzymes to enhance salt tolerance [42]. Transgenic potato overexpressing the *StERF94* gene could improve salinity resistance by increasing antioxidant activity [43]. Similarly, as a major osmotic regulator, CAT can also effectively reduce excess ROS in plants [44]. Overexpression of wheat *TaCAT-3B* in *Escherichia coli* enhanced resistance to abiotic stress, such as heat, drought, salt and various concentrations of arsenic (As) [45]. Therefore, the activities of POD, SOD and CAT are important parameters in plant response to high salinity. In this study, we measured the physiological and biochemical indices of transgenic poplars under salt stress. No significant differences were observed between transgenic poplars and WT plants under control conditions. However, the activities of POD, SOD and CAT of transgenic poplars were significantly higher than those of WT plants under salt stress, whereas MDA content showed an opposite trend. In addition, under salt stress, the expression levels of POD- and SOD-related genes were significantly increased in transgenic poplars compared with WT plants. The results indicate that the *PagERF072* gene regulates antioxidant enzyme activities to increase salt tolerance.

## 4. Materials and Methods

### 4.1. Plant Materials

84K poplars (*Populus alba* × *P. glandulosa*) were used as plant materials in our study and were planted in the experimental field of Northeast Forestry University, Harbin, China. The poplar plants used in the experiments were propagated by tissue culture. Poplar leaves were placed on callus induction medium to obtain callus. Next, the proliferated calluses were transplanted to regeneration medium to obtain shoots; then, shoots were transplanted to rooting medium. Callus induction Murashige and Skoog (MS) medium consists of 0.1 mg/mL NAA and 0.04 mg/mL thidiazuron, and regeneration 1/2 MS medium consists of 0.01 mg/mL 1-naphthaleneacetic acid (NAA) and 0.1 mg/mL indole-3-butytric acid (IBA). One-month tissue-cultured seedlings were transplanted into mixed soil containing vermiculite and perlite (5:3:2, *v*/*v*/*v*). Then, the plants were grown in a greenhouse at 25 °C with 16 h light/8 h dark cycles.

One-month poplar seedlings cultured in soil were irrigated with 150 mM NaCl or 50 µM abscisic acid (ABA) for 0, 3, 6, 12, 24 and 48 h. The primary leaves, transition leaves, secondary leaves, primary stems, secondary stems and roots were sampled at each time point and stored at −80 °C. Each sample contained three biological replicates.

Tobacco seeds were spread evenly in flower pots with the mixed soil containing vermiculite and perlite (5:3:2, *v*/*v*/*v*), and one-month seedlings were used for transformation.

### 4.2. Gene Screening and Sequence Analysis

The fragments per kilobase of exon model per million mapped fragments (FPKM) abundance of 174 ERF family genes was extracted from previous research [25]. Differentially expressed genes (DEGs) among the 174 ERF family genes were screened by the DESeq2 package [46] with the standards of Log2 (fold change) ≥ 1 and padj ≤ 0.05. The ERF family genes were named with reference to previous studies [47].

The sequence information of the *PagERF072* (homologous gene of *Populus trichocarpa Potri.010G006800.1*) gene was obtained from the *Populus trichocarpa* v3.0 database in Phytozome (https://phytozome.jgi.doe.gov/pz/portal.html, accessed on 20 November 2020). Total RNA was extracted using an RNA extraction kit (TAKARA, Dalian, China) and reversed to cDNA using a reverse transcription kit (TAKARA, Dalian, China). The transcript sequence of the *PagERF072* gene was cloned by nested PCR. An ABI7500 real-time system was used for qRT-PCR. A TB Green^®^ Premix Ex Taq™ II kit (TAKARA, Dalian, China) was used for qRT-PCR. The procedure described in our previous study [48]. Relative expression levels of genes were calculated by 2^−ΔΔCt^ method [49], and *actin* was used as a reference gene in qRT-PCR experiments. All primers are listed in Appendix A.

### 4.3. Phylogenetic Analysis of PagERF072 Protein

Homologous protein sequences of PagERF072 were obtained from NCBI (http://www.ncbi.nlm.nih.gov/, accessed on 20 November 2020). Multiple alignments of protein sequences from different species were compared by ClustalW [50]. The phylogenetic tree of homologous proteins was constructed by MEGA6 using the neighbor-joining method [51].

### 4.4. Subcellular Localization of PagERF072

The ORF sequence of the *PagERF072* gene without a stop codon was ligated to the pBI121-GFP vector, which was driven by CaMV35S promoter. Then, the fusion plasmid was transformed into GV3101 *Agrobacterium*. One-month-old *N. benthamiana* seedlings were used for transient transformation. *Agrobacterium* solution containing 35S-GFP and 35S-PagERF072-GFP was injected into the tobacco inner epidermis and cultured in the dark for 24 h. Fluorescence signal was observed by laser confocal scanning microscopy (LSM 800, Zeiss, Jena, Germany).

### 4.5. Transcription Activation Assay of PagERF072

The CDS sequence of the *PagERF072* gene was cloned from 84K cDNA using specific primers and ligated to the pGBKT7 vector to form a pGBKT7-ERF072 fusion vector. The pGBKT7-ERF072 fusion vectors, pGBKT7 vector (negative control) and pGBKT7-53/pGADT7-T (positive control) were transformed into Y2H yeast strains. The positive yeast strains were screened on SD/-Trp medium, and β-galactosidase activity was verified on SD/-Trp/-his/X-α-Gal medium.

The GCC-box element with three tandem repeats was inserted into the pAbAi vector to form a bait reporter vector. The CDS sequence of the *PagERF072* gene was inserted into the pGADT7 vector to form a pGADT7-ERF072 fusion vector. The bait reporter vector was transformed into Y1H yeast by a transformation system. The pGADT7-ERF072 vector was transformed into positive bait reporter yeast, which was then spread on SD/-Leu and SD/-Leu/AbA medium with dilution 10, 100 and 1000 times for verification.

The GCC-box element with three tandem repeats was inserted into the pCAMBIA1301 vector [52] to form a reporter vector, and the full-length *PagERF072* sequence was inserted into the pROK2 vector to form a pROK2-ERF072 effector vector. An *Agrobacterium*-mediated method was used to co-transform effector vectors and reporter vectors into tobacco leaves. The reporter vector and empty pROK2 vector were co-transformed into tobacco leaves as control. The procedure of GUS staining was described as previous research [48]. First, the tobacco leaves were vacuumed in GUS staining solution for 20 min and kept overnight at 37 °C in the dark. Then, the tobacco leaves were decolorized with decolorizing solution and photographed for observation.

### 4.6. Generation of Transgenic Poplar Lines

The CDS sequence of *PagERF072* was inserted into the pBI-121 vector with a *SpeI* restriction site to construct a pBI-121-ERF072 fusion vector. The fusion vector was transformed into *Agrobacterium* GV3101. The leaves from one-month-old tissue-cultured seedlings were cut into discs, and the leaf discs were immersed in *Agrobacterium* GV3101 solution with a concentration of OD_600_ = 0.6 for 10 min. Then, the leaf discs were spread on MS differentiation medium and cultured in the dark for 3 days. Then, the leaf discs were transferred to MS selection medium containing 50 mg/L kanamycin and 200 mg/L cephalosporin. Once poplar buds grew, they were transplanted into MS rooting medium containing 50 mg/L kanamycin and 200 mg/L cephalosporin for one month. Finally, positive transgenic poplars were screened by PCR and RT-PCR using specific primers.

### 4.7. Morphological and Physiological Measurements

To explore the biological function of the *PagERF072* gene, the tissue-cultured transgenic poplars and WT control plants were transplanted to soil. The one-month-old plants were irrigated with 0 mM and 150 mM NaCl for five days, respectively. After measurement of morphological traits, leaves were harvested from the tops of the plants for determination of physiological indicators. The activities of CAT, POD and SOD, as well as the content of MDA, were measured by Suzhou Comin Biotechnology (Suzhou, China) (www.cominbio.com, accessed on 20 February 2020) (test kit numbers CAT-1-Y, POD-1-Y, SOD-1-Y and MDA-1-Y, respectively).

### 4.8. Histochemical Staining

One-month-old transgenic poplars and WT control plants were used for histochemical staining analysis under salt stress. The activities of hydrogen peroxide and superoxide in plants were detected using 3, 3′-diaminobenzidine (DAB) and nitrotetrazolium blue chloride (NBT) staining [53]. The collected leaves were immersed in staining solution, stained at 37 °C for 12 h, decolorized with ethanol and photographed for observation.

## 5. Conclusions

In this study, we cloned a significantly salt-induced ERF gene, *PagERF072*, which was localized in the nucleus and exhibited no transactivation. The results of Y1H confirmed that the PagERF072 protein can specifically bind to GCC-box element. Subsequently, we obtained three transgenic poplar lines overexpressing *PagERF072* by tree genetic transformation, which displayed growth advantage relative to WT plant at morphological and physiological levels under salt stress. Furthermore, *PagERF072* can respond to salt stress by regulating the expression of POD- and SOD-related genes. Further experiments are required to verify these results. In addition, we found that the *PagERF072* gene can positively respond to ABA stress, with increased expression levels following exposure to 50 µM ABA; however, whether *PagERF072* is involved in the ABA signaling pathway is still uncertain. In conclusion, *PagERF072* functions as a positive regulator of plant salt response. The results of the present study provide insights into the molecular mechanism of ERF genes with respect to salt tolerance.

## Figures and Tables

**Figure 1 ijms-23-10707-f001:**
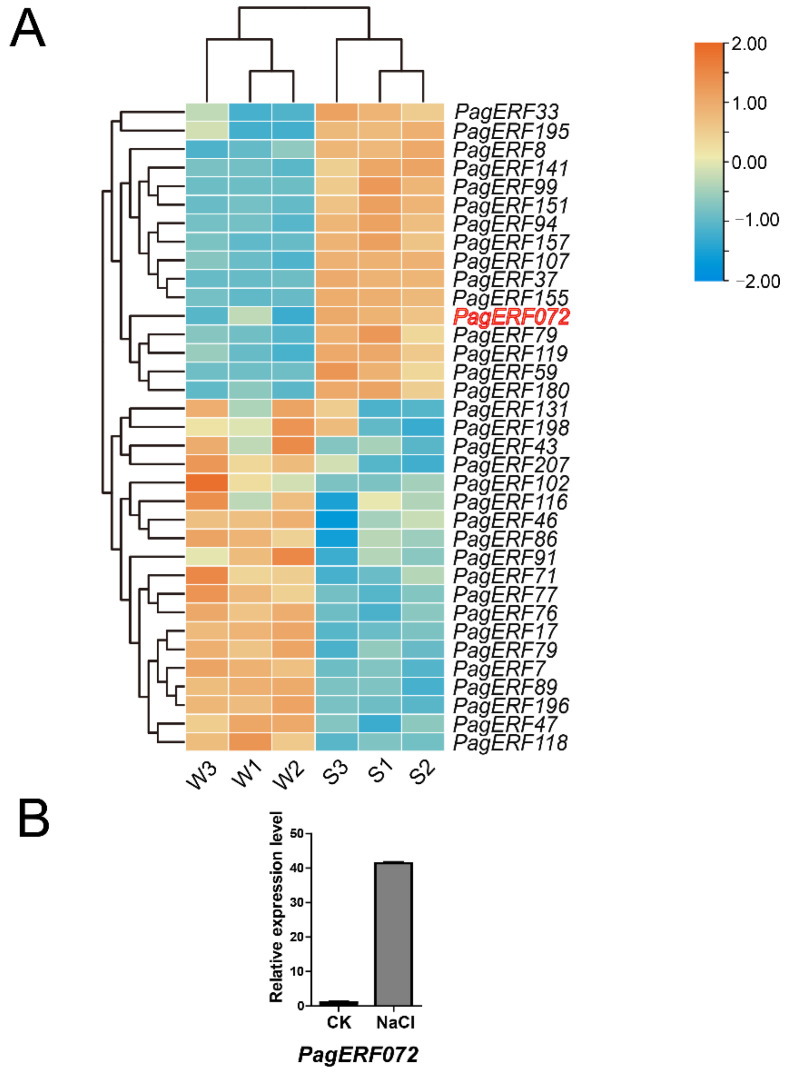
Heatmap of differentially expressed ERF genes after 0 and 150 mM NaCl treatment. (**A**) Orange color represents high expression; blue color represents low expression; W1–3: control condition; S1–3: 150 mM NaCl treatment. (**B**) Gene expression analysis by qRT-PCR. CK and NaCl represent control condition and salt treatment, respectively.

**Figure 2 ijms-23-10707-f002:**
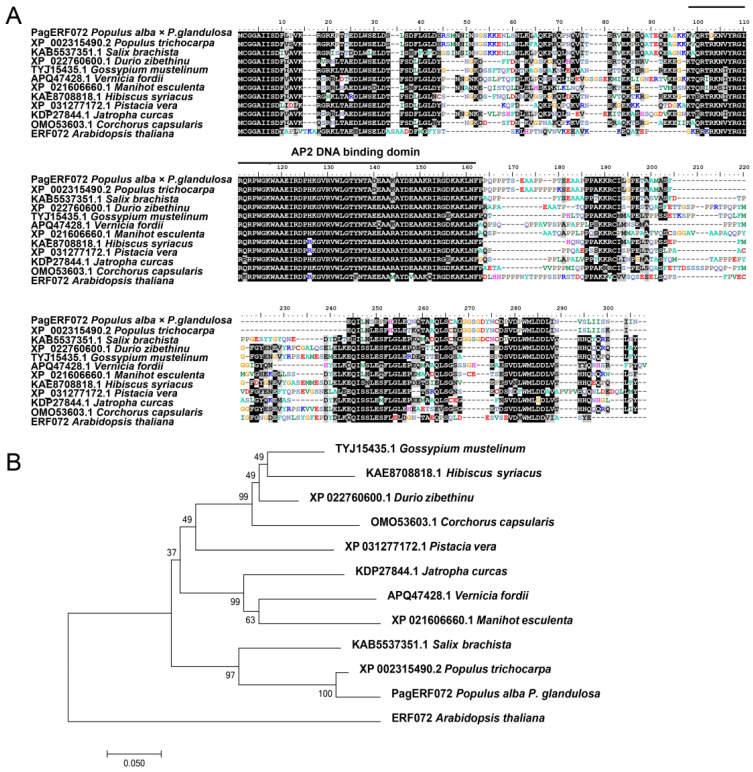
Sequence analysis of PagERF072 protein. (**A**) Protein alignment of PagERF072 and its homologous proteins from other plant species. (**B**) Phylogenetic analysis between PagERF072 and other ERF proteins. Multiple alignments of protein sequences were compared by ClustalW. The phylogenetic tree of homologous proteins was constructed by MEGA6 using the neighbor-joining method with a bootstrap value of 1000 replicates.

**Figure 3 ijms-23-10707-f003:**
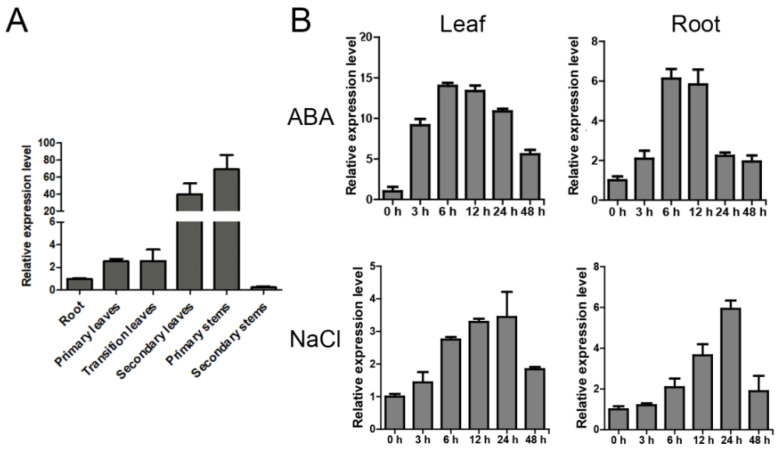
Expression patterns of the *PagERF072* gene in leaves and roots under salt stress and ABA treatments. (**A**) Expression levels of the *PagERF072* gene in different tissues. (**B**) Expression patterns of *PagERF072* in response to NaCl and ABA. Data were processed using the 2^−ΔΔCt^ method. Error bars represent standard deviation.

**Figure 4 ijms-23-10707-f004:**
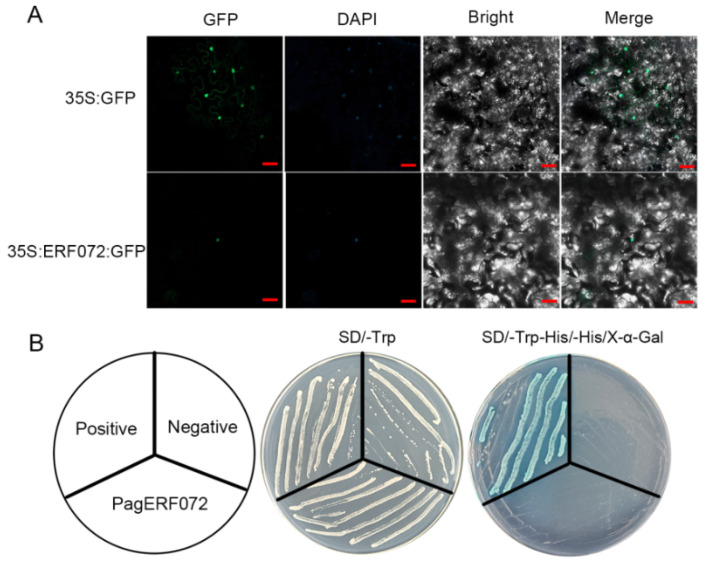
Subcellular localization and transcriptional activation of PagERF072. (**A**) Subcellular localization of PagERF072 protein. Scale bar = 20 µm. (**B**) Transcriptional activation of PagERF072 protein.

**Figure 5 ijms-23-10707-f005:**
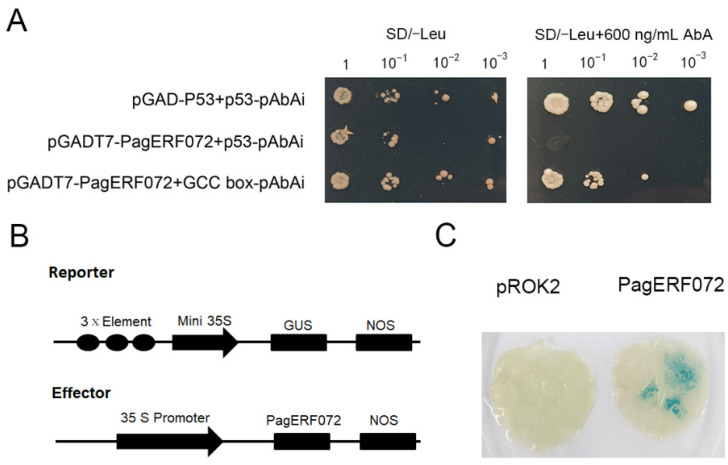
Specific binding element of PagERF072. (**A**) Specific binding of PagERF072 to GCC-box. (**B**) Schematic diagrams of the effector and reporter vectors used for coexpression in tobacco. (**C**) GUS staining.

**Figure 6 ijms-23-10707-f006:**
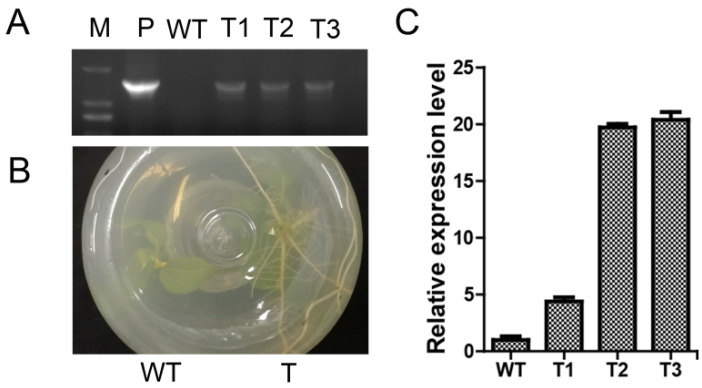
Identification of positive transgenic poplar lines. WT, wild type; T1–3, transgenic poplar lines overexpressing *PagERF072*. (**A**) PCR validation of transgenic poplar lines at the DNA level; M, 2000 DNA marker; P, positive plasmid. (**B**) Morphological phenotype of transgenic poplar; T, transgenic poplar. (**C**) qRT-PCR validation of transgenic poplar lines.

**Figure 7 ijms-23-10707-f007:**
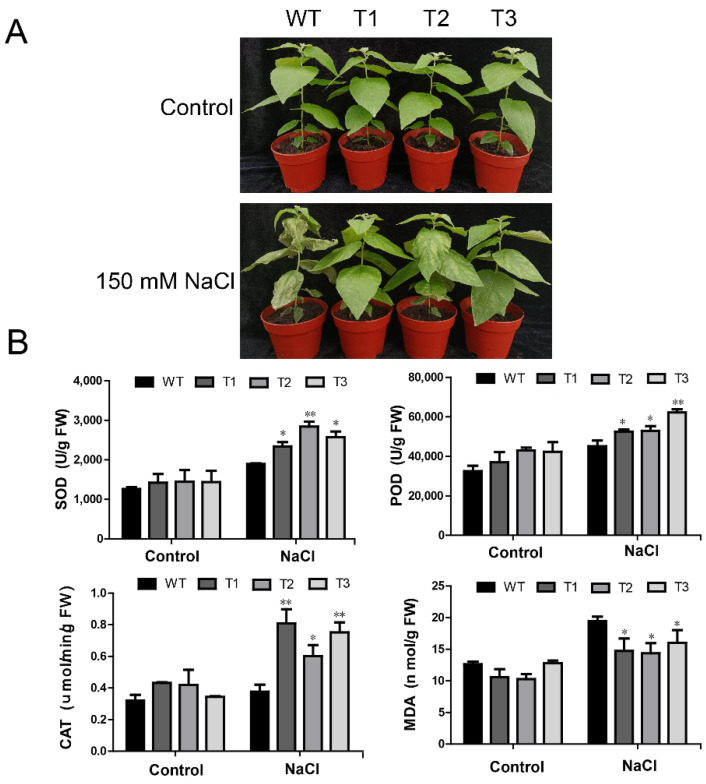
Morphological and physiological characteristics of transgenic poplars under salt stress. (**A**) Morphological phenotypes of transgenic poplars. (**B**) Physiological indices of transgenic poplars; FW, fresh weight. Error bars indicate mean ± SD. Asterisks indicate significant differences between transgenic lines and wild-type lines (*t* test, * *p* < 0.05; ** *p* < 0.01).

**Figure 8 ijms-23-10707-f008:**
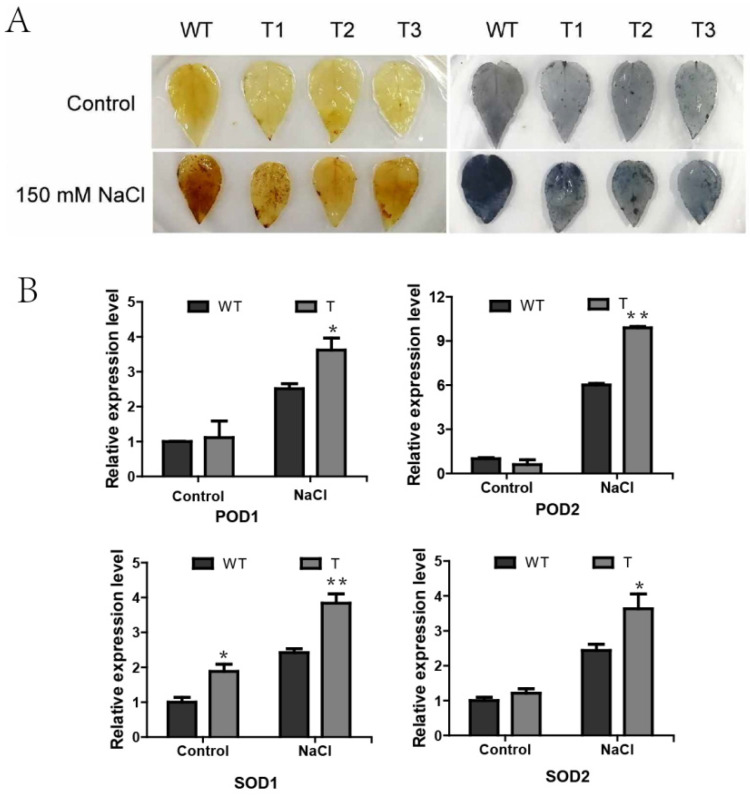
ROS scavenging assay. (**A**) NBT and DAB staining were conducted to detect the content of hydrogen peroxide; WT, wild-type plants; T1–3, transgenic poplar lines overexpressing *PagERF072.* (**B**) Expression analysis of SODs and PODs. WT, wild-type plants; T, transgenic poplar. Asterisks indicate significant differences between transgenic lines and wild-type lines (*t* test, * *p* < 0.05; ** *p* < 0.01).

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
