# Peer review of "Overexpression of PagERF072 from Poplar Improves Salt Tolerance"

_ijms, 2022, doi:10.3390/ijms231810707_

Round 1

Reviewer 1 Report (Previous Reviewer 2)

I have already reviewed this Ms earlier two times and suggested publication after some minor revision. 

Author Response

Thank you for your comment,we have revised the manuscript.

Reviewer 2 Report (Previous Reviewer 1)

30. I am simply asking whether the ERF072 gene sequence used in this experiment was from Populus alba or P. glandulosa. It is clear that all poplars, including Populus trichocarpa, have the homologous sequence, but not all ERF072 have the same function as PagERF072 used in this experiment. Therefore, it is necessary to clarify which of the two poplar species PagERF072 is derived from.

31. The authors do not understand the meaning of the question and therefore do not give an appropriate reply. Does the ‘0’ mean the average of FPKM values of all genes?

32. The description of how poplar is propagated by tissue culture is confusing and needs revision. For example, ‘The poplar plants used in the experiments were propagated by tissue culture. Poplar explants (leaves?, stems?) were placed on the callus induction medium to obtain callus. Next, the proliferated callus were transplanted to the regeneration medium to obtain shoots, and then shoots were transplanted to the rooting medium. The callus induction medium consists of …, the regeneration medium consists of ….’

Author Response

Suggestions for Authors

  1. I am simply asking whether the ERF072 gene sequence used in this experiment was from Populus alba or P. glandulosa. It is clear that all poplars, including Populus trichocarpa, have the homologous sequence, but not all ERF072 have the same function as PagERF072 used in this experiment. Therefore, it is necessary to clarify which of the two poplar species PagERF072 is derived from.

——Response: The 84K is derived from the hybrid offspring of Populus alba and Populus glandulosa, 84K is a new variety and can be genetically stable. Poplar 84K is a fast-growing poplar hybrid. Originated in South Korea, this hybrid has been extensively cultivated in northern China. And its genome has published in DNA Research [1]. In this study, 84K is the experiment material, and the PagERF072 gene sequence was obtained from 84K poplar genome.

  1. Qiu D, Bai S, Ma J, Zhang L, Shao F, Zhang K, Yang Y, Sun T, Huang J, Zhou Y et al: The genome of Populus alba x Populus tremula var. glandulosa clone 84K. DNA research : an international journal for rapid publication of reports on genes and genomes 2019, 26(5):423-431.

  1. The authors do not understand the meaning of the question and therefore do not give an appropriate reply. Does the ‘0’ mean the average of FPKM values of all genes?

——Response: Thank you for your comment. The “0” mean the average of FPKM values of all genes. We used the software to perform standard normal distribution (Z-score) of the FPKM values. The FPKM value was normalized by log2, which conforms to a standard normal distribution with average of 0 and a variance of 1. If the original value is lower than Z, it is negative, otherwise it is positive.

  1. The description of how poplar is propagated by tissue culture is confusing and needs revision. For example, ‘The poplar plants used in the experiments were propagated by tissue culture. Poplar explants (leaves?, stems?) were placed on the callus induction medium to obtain callus. Next, the proliferated callus were transplanted to the regeneration medium to obtain shoots, and then shoots were transplanted to the rooting medium. The callus induction medium consists of …, the regeneration medium consists of ….’

——Response: Thank you for your comment. We have modified the whole sentence as “The poplar plants used in the experiments were propagated by tissue culture. Poplar leaves were placed on the callus induction medium to obtain callus. Next, the proliferated calluses were transplanted to the regeneration medium to obtain shoots, and then shoots were transplanted to the rooting medium. The callus induction Murashige and Skoog (MS) medium consists of 0.1 mg/mL NAA and 0.04 mg/mL thidiazuron, and the regeneration 1/2 MS medium consists of 0.01 mg/mL 1-Naphthaleneacetic acid (NAA) and 0.1 mg/mL Indole-3-Butytric acid (IBA)”.

Round 2

Reviewer 2 Report (Previous Reviewer 1)

In 4.1, delete '. Plant cells went a process of de-differentiation and re-differentiation to produce the complete plants seedlings. ' .

Author Response

In 4.1, delete '. Plant cells went a process of de-differentiation and re-differentiation to produce the complete plants seedlings. ' .

——Response:  Thank you for your comment. And we have deleted the sentence.

This manuscript is a resubmission of an earlier submission. The following is a list of the peer review reports and author responses from that submission.

Round 1

Reviewer 1 Report

This manuscript reports that overexpression of PagERF072 enhances salt tolerance of transgenic poplar by upregulating the gene for ROS-scavenging enzymes. Some data are interesting and may provide insights into the the functions of the ERF transcription factor; however, the overall results are not novel, and the manuscript contains several deficiencies. English expression needs to be corrected. Therefore, I am sorry to say that I cannot recommend this form of the manuscript for publication in International Journal of Molecular Sciences.

Basically, due to the lack of novelty in the results, this MS is not suitable for publication in high impact factor journals.

The name of the protein PagERF072 is inappropriate. It should be easy to find out from which poplar the ERF072 is derived, P. alba or P. glandulosa.

Please indicate why PagERF072 is analyzed in detail in this study though the other genes are also highly salt-induced.

L55. In this study or [25]?

L66. What 84k means?

The resalt in Fig.1 is very unusual. The expressions of about half of the ERF gene are suppressed by water treatment, and the expression of the remaining half of the genes are enhanced by water treatment. The expression levels of most genes should not change (around 0.00) with water treatment. These data reduce the reliability of this experiment.

In the legends to Fig.1. Gene expression analysis by RNA-seq and RT-PCR. Which is correct? In Fig.1A, gene names should be in italic font.

Fig.2A is too small.

In Fig2.B, please include AA sequences of some PagERF proteins and well-characterize ERF proteins from other species, such as AtERF98. Because these are AA sequences, the accession numbers should be in normal font.

Fig.3 is too small. Please put A and B side by side.

Primary and secondary xylem tissues are very specific. Please describe the reason why those specific tissues were used for analysis and how to prepare RNA from those specific tissues.

L104. PBI121 should be pBI121.

The contrast in Fig 4A is too low.

No scale bar.

L124. with dilution of 0, 10, 100 and 100 times?

L139 and 141. Please check Figure6A and 6B are correct.

In Fig.6B legends, a photo of culture medium is not ‘Morphological phenotype’. No ‘OE’.

Please unify the font size of figure legends.

L178. No statistical analysis.

Why SOD activities in transgenic plants are not enhanced in Fig.7B but the gene expression of SOD1 was up-regulated in Fig8B?

Why were CAT gene expressions not examined?

Why ROS-scavenging enzyme activities and/or gene expressions were not enhanced in transgenic plants under control condition although the ERF gene is controlled by the constitutive 35S promoter? This is unusual. Normally, constitutive over-expression of TFs resulted in enhanced ROS-scavenging enzyme gene expressions under non-stressed condition. For example, please see [20] etc..

[33] is wrong reference. Please check all references.

L228. Plant tolerance?

The description of Method is insufficient.

Which poplar species is used for production of transgenic plants. Which gene is used as a reference gene in qRT-PCR experiments?

L241. Why NAA/IBA or NAA/TDZ were used for seedling cultivation? This is for callus formation?

L234 and L246 Both are ‘one-month-old’?

L246. Both salt and ABA were added?

L250. Soil with soil?

English correction is required throughout the manuscript.

For example, in L59, ‘The study provides a foundation for molecular function of PagERF072 gene’ should be reconsidered.  

L253. ‘The mRNA abundance of 174 ERF family genes was found from previous research’ is unclear.

TAKALA should be TAKARA.

L281. ‘PagERF072 gene was cloned by specific primer pairs’ is not appropriate.

L303. ‘ligated to PBI-121 vector using specific primers’ is not appropriate.

L318. MDA is not enzyme, so does not have activity.

Author Response

This manuscript reports that overexpression of PagERF072 enhances salt tolerance of transgenic poplar by upregulating the gene for ROS-scavenging enzymes. Some data are interesting and may provide insights into the the functions of the ERF transcription factor; however, the overall results are not novel, and the manuscript contains several deficiencies. English expression needs to be corrected. Therefore, I am sorry to say that I cannot recommend this form of the manuscript for publication in International Journal of Molecular Sciences.

 Basically, due to the lack of novelty in the results, this MS is not suitable for publication in high impact factor journals.

——Response: Thanks a lot for your proposal, in this study, We explored the molecular function of the PagERF072 gene verified by phylogenetic tree, transcriptional activation, subcellular localization, and Y1H experiments, in addition, we obtained PagERF072 overexpressing transgenic lines, and through physiological and biochemical experiments proved that PagERF072 could increase the salt stress tolerance of transgenic poplars by regulating the expression of POD and SOD-related genes. This study will lay a theoretical foundation for further study on molecular function of PagERF072 in poplar. Therefore, we have made a comprehensive revision of the paper and hope that the reviewers will consider our manuscript again

  1. The name of the protein PagERF072 is inappropriate. It should be easy to find out from which poplar the ERF072 is derived, P. alba or P. glandulosa.

——Response: In our study, we used 84K poplar as the experimental material, ‘Pag’ is the abbreviation of 84K, which has also been reported in other articles (KNAT2/6b, a class I KNOX gene, impedes xylem differentiation by regulating NAC domain transcription factors in poplar) or (The PagKNAT2/6b-PagBOP1/2a Regulatory Module Controls Leaf Morphogenesis in Populus), the homologous gene of PagERF072 in Arabidopsis is ERF72, and we named it as PagERF072.

  1. Please indicate why PagERF072 is analyzed in detail in this study though the other genes are also highly salt-induced.

——Response: In our experiment, we found several up-regulated ERF family genes in the RNA-Seq of salt stress. We found that ERF72 of Arabidopsis thaliana is involved in abiotic stress. Therefore, we chose PagERF72 as our research object. In addition, we are also studying other genes, and we will publish our results in the future.

  1. L66. What 84k means?

——Response: 84K is the abbreviation of Populus alba × P. glandulosa, and we have mentioned it in the manuscript.

  1. The resalt in Fig.1 is very unusual. The expressions of about half of the ERF gene are suppressed by water treatment, and the expression of the remaining half of the genes are enhanced by water treatment. The expression levels of most genes should not change (around 0.00) with water treatment. These data reduce the reliability of this experiment.

——Response: In our study, we used 150 mM NaCl for stress treatment, and normal watering was used as a control. The only variable in the experiment was the salt treatment, and there were no other variables, so our experiment was accurate.

  1. In the legends to Fig.1. Gene expression analysis by RNA-seq and RT-PCR. Which is correct? In Fig.1A, gene names should be in italic font.

——Response: qRT-PCR is correct and we have deleted RNA-seq in the legend of Fig 1. We have changed gene names to italics.

  1. Fig.2A is too small. In Fig2.B, please include AA sequences of some PagERF proteins and well-characterize ERF proteins from other species, such as AtERF98. Because these are AA sequences, the accession numbers should be in normal font.

 ——Response: We have modified the resolution of the image. In this study, we performed BLAST on the PagERF72 protein through the NCBI online website, and selected 10 proteins from other species including Populus trichocarpa, Salix brachista, Durio zibethinus, Gossypium mustelinum, Vernicia fordii, Manihot esculenta, Hibiscus syriacus, Pistacia vera, Corchorus capsularis and Jatropha curcas, these are high homolog proteins of PagERF72, and according to the phylogenetic tree of 11 proteins, these proteins have relatively close relationship. For the figure 2, we have modified the font of the accession numbers.

  1. Fig.3 is too small. Please put A and B side by side. Primary and secondary xylem tissues are very specific. Please describe the reason why those specific tissues were used for analysis and how to prepare RNA from those specific tissues.

——Response: Thank you for your suggestion, we have modified the resolution of the image, and swap the positions of A and B. We also describe the method of the sample collection in detail in Materials and methods.

  1. L104. PBI121 should be pBI121.

——Response: Thanks, we have changed PBI121 to pBI121.

  1. The contrast in Fig 4A is too low. No scale bar.

——Response: We have modified the resolution of the image, and added scale bar in figure 4A.

  1. L124. with dilution of 0, 10, 100 and 100 times?

——Response: We have deleted the sentence “with dilution of 0, 10, 100 and 100 times” in the manuscript. 

  1. L139 and 141. Please check Figure6A and 6B are correct. In Fig.6B legends, a photo of culture medium is not ‘Morphological phenotype’. No ‘OE’. Please unify the font size of figure legends.  

——Response: Sorry for the mistakes, we have modified the figure 6 and changed OE to T, and also adjusted the font size of figure legends.

  1. L178. No statistical analysis.

——Response: Thank you for your comment, we did statistical analysis and added the stars in the figure 8.

  1. Why SOD activities in transgenic plants are not enhanced in Fig.7B but the gene expression of SOD1 was up-regulated in Fig8B? Why were CAT gene expressions not examined?

——Response: We observed that SOD1 activity was also increased in Fig.7B, but the difference was not significant, while in Fig.8B, what we showed is the relative expression level of SOD1. We picked several CAT genes, but we found no difference after stress, therefore, we did not add. There may be other CAT genes that function.

  1. Why ROS-scavenging enzyme activities and/or gene expressions were not enhanced in transgenic plants under control condition although the ERF gene is controlled by the constitutive 35S promoter? This is unusual. Normally, constitutive over-expression of TFs resulted in enhanced ROS-scavenging enzyme gene expressions under non-stressed condition. For example, please see [20] etc..

——Response: We believe that under normal conditions, PagERF72 does not activate the expression of POD, SOD-related genes, but under the stimulation of salt stress, PagERF72 can activate the expression of these genes, thereby participating in salt stress, our results are similar to other genes, such as BpERF13 [1], SlERF84 [2] and OsERF71 [3].

[1]    Lv, K.;Li, J.;Zhao, K.;Chen, S.;Nie, J.;Zhang, W.;Liu, G.;Wei, H. Overexpression of an AP2/ERF family gene, BpERF13, in birch enhances cold tolerance through upregulating CBF genes and mitigating reactive oxygen species [J]. Plant Science, 2020, 292

[2]    Li, Z.;Tian, Y.;Xu, J.;Fu, X.;Gao, J.;Wang, B.;Han, H.;Wang, L.;Peng, R.;Yao, Q. A tomato ERF transcription factor, SlERF84, confers enhanced tolerance to drought and salt stress but negatively regulates immunity against Pseudomonas syringae pv. tomato DC3000 [J]. Plant physiology and biochemistry, 2018, 132(683-695).

[3]    Li, J.;Guo, X.;Zhang, M.;Wang, X.;Zhao, Y.;Yin, Z.;Zhang, Z.;Wang, Y.;Xiong, H.;Zhang, H. OsERF71 confers drought tolerance via modulating ABA signaling and proline biosynthesis [J]. Plant Science, 2018, 270(131-139).

  1. [33] is wrong reference. Please check all references.

——Response: Sorry for the mistakes, we have corrected the reference 33 and double checked all references.

  1. L228. Plant tolerance?

——Response: We have modified the whole sentence as “Therefore, the activities of POD, SOD and CAT are important parameters in plant response to high salinity”.

The description of Method is insufficient.

  1. Which poplar species is used for production of transgenic plants. Which gene is used as a reference gene in qRT-PCR experiments?

——Response: The 84K (Populus alba × P. glandulosa) was used for production of transgenic plants, and the Actin is used as a reference gene in qRT-PCR experiments. We have added the sentence “the Actin was used as a reference gene in qRT-PCR experiments” in the manuscript.

  1. L241. Why NAA/IBA or NAA/TDZ were used for seedling cultivation? This is for callus formation?

——Response: Thank you for your comment. IBA can promote the generation of adventitious roots, and NAA/IBA was used for rooting of resistant seedlings. TDZ can promote cell division and differentiation, and NAA/TDZ was used for callus formation of resistant seedlings.

  1. L234 and L246 Both are ‘one-month-old’?

——Response: We have modified to one month old.

  1. L246. Both salt and ABA were added?

——Response: We just treated the plants with salt or ABA separately, and we have changed the sentence as follows: One month old poplar seedlings cultured in the soil were irrigated with 150 mM NaCl or 50 µM abscisic acid (ABA) for 0, 3, 6, 12, 24, and 48 hours, respectively.

  1. L250. Soil with soil?

——Response: We have modified the whole sentence as “One month tissue-cultured seedlings were transplanted into the mixed soil containing vermiculite and perlite 5:3:2 (v/v/v)”

English correction is required throughout the manuscript.

  1. For example, in L59, ‘The study provides a foundation for molecular function of PagERF072 gene’ should be reconsidered.  

——Response: We have modified the whole sentence as “The study identifies the molecular function of PagERF072 gene, which provides a potential contribution in salt-tolerance of poplar.”

  1. L253. ‘The mRNA abundance of 174 ERF family genes was found from previous research’ is unclear.

——Response: We have modified the whole sentence as “The Fragments Per Kilobase of exon model per Million mapped fragments (FPKM) abundance of 174 ERF family genes was found from previous research.”

  1. TAKALA should be TAKARA.

——Response: We have changed TAKALA to TAKARA.

  1. L281. ‘PagERF072 gene was cloned by specific primer pairs’ is not appropriate.

——Response: We have modified the whole sentence as “The CDS sequence of PagERF072 gene was cloned from 84K cDNA using specific primers”

  1. L303. ‘ligated to PBI-121 vector using specific primers’ is not appropriate.

——Response: We have modified the whole sentence as “The CDS sequence of PagERF072 was inserted into pBI-121 vector to construct pBI-121-ERF072 fusion vector.”

  1. L318. MDA is not enzyme, so does not have activity.

——Response: We have changed activity to content.

Reviewer 2 Report

The Ms entitled “Overexpression of PagERF072 from poplar improves salt tolerance” is very nicely designed and executed.

The Ms is also very well written. It may be accepted for publication after suggested changes-

1.      Abstract should be more elaborated.

2.      68.  Amino acid residues.

3.      How many positive events were identified?

4.      Author should try to confirm the number of gene integration in the transgenic lines.

5.      Discussion should be more elaborated. Author may also discuss these Ms about role of SOD and Catalase in discussion. https://www.sciencedirect.com/science/article/abs/pii/S0304389420315715

https://link.springer.com/chapter/10.1007/978-981-15-0690-1_3

6.      319, link not working. Author may mention the name of kit here.

7.      Conclusions need to elaborated. It seems to repetition of abstract. Author may include application and future perspectives also.

Figure 8, statistical analysis should be done.

8.      Language is OK, some minor spell check and typos needs to be checked. 

Author Response

Thank you for your comment, we have responded all the questions one by one in the below files.

Round 2

Reviewer 1 Report

This MS does not have novelty, so is not suitable for publication in IJMS. It is well known that ERF TFs control plant stress tolerance, by regulating ROS-scavenging enzyme expressions/activities.

Please clarify which poplar the gene is from, or whether the sequence is present in both poplars.

The result in Fig.1 is very unusual and unreliable.

The expressions of about half of the ERF gene are suppressed by water treatment, and the expression of the remaining half of the genes are enhanced by water treatment. The expression levels of most genes should not change (around 0.00) with water treatment.

 In Fig2.B, In phylogenetic analysis, the diversity of the proteins cannot be understood unless some distantly related proteins are added in addition to closely related proteins.

-Response: Thank you for your comment. IBA can promote the generation of adventitious roots, and NAA/IBA was used for rooting of resistant seedlings. TDZ can promote cell division and differentiation, and NAA/TDZ was used for callus formation of resistant seedlings.

>Please explain in the text why the authors needed to promote cell division, differentiation, and callus formation of resistant seedlings.

—Response: We have modified the whole sentence as “The CDS sequence of PagERF072 was inserted into pBI-121 vector to construct pBI-121-ERF072 fusion vector.”

>Please clarify the restriction enzyme used.

Author Response

1  The expressions of about half of the ERF gene are suppressed by water treatment, and the expression of the remaining half of the genes are enhanced by water treatment. The expression levels of most genes should not change (around 0.00) with water treatment.

——Response: In our study, RNA-Seq was performed using salt solution-stressed poplars, and under normal-growth poplars were used as controls. We did not perform water treatment, and watering was only for the normal growth of plants. This treatment is also reflected in other species, such as Cucumis melo L (Genome-wide characterization of MATE family members in Cucumis melo L. and their expression profiles in response to abiotic and biotic stress), Populus talassica × Populus euphratica (Transcriptomic Profile Analysis of Populus talassica × Populus euphratica Response and Tolerance under Salt Stress Conditions) and Ipomoea batatas L (Genome-wide survey and expression analysis of GRAS transcription factor family in sweetpotato provides insights into their potential roles in stress response).

2  In Fig2.B, In phylogenetic analysis, the diversity of the proteins cannot be understood unless some distantly related proteins are added in addition to closely related proteins.

——Response: We have added Arabidopsis and performed Blast analysis, we found that the sequence similarity between Arabidopsis ERF072 protein and PagERF072 protein was 43.33%.

3 >Please explain in the text why the authors needed to promote cell division, differentiation, and callus formation of resistant seedlings.

——Response:We have explained in the manuscript,and we have changed the sentence as follows: The tissue cell division and differentiation were grown on with MS medium containing 0.1 mg/mL NAA and 0.04 mg/mL thidiazuron. Plant cells undergo a process of de-differentiation and re-differentiation to eventually produce resistant seedlings.

4 >Please clarify the restriction enzyme used.

We have added.

Reviewer 2 Report

The reference list is not updated as per the revised Ms. I could see 53 cited references in Ms, but only 51 in the list. 

Author Response

1 参考文献列表没有按照修改后的 Ms 更新。我可以看到 Ms 引用的参考文献有 53 篇,但列表中只有 51 篇。 

——回应:感谢您的评论,我们已更新所有参考资料。

Round 3

Reviewer 1 Report

Again, I have to make the same comment as last time. This MS does not have novelty, so is not suitable for publication in IJMS. It is well known that ERF TFs control plant stress tolerance, by regulating ROS-scavenging enzyme expressions/activities.

The authors ignored my question. So, I would like to ask the same question again. Please clarify which poplar the gene is from, or whether the sequence is present in both poplars.

In Fig.1, it is curious that many TF genes are divided into only two expression levels and expression patterns.

Please define the expression level ‘0’ as a standard or show fold changes of the gene expression.

What ‘W (W1-3)’ in Fig1 represent for?

Though the authors explain why the authors needed to promote cell division, differentiation, and callus formation of resistant seedlings, I donot understand the reason. English of these sentences also needs to be reconsidered.